**Comment**

# Rethinking misinformation through plausibility estimation and confidence calibration

Valentin Guigon, Lucille Geay & Caroline J. Charpentier

Democracies are vulnerable to misinformation. Prevailing interventions emphasize truth detection, but offer no panacea. We argue that strengthening people's ability to estimate the plausibility of information and calibrate their confidence under uncertainty offers a complementary route to addressing misinformation.

Public and private initiatives are investing heavily in combating misinformation by preparing people through education programs, curbing circulation with platform-level friction, and correcting false claims after exposure. Each approach delivers specific benefits. Critical-thinking education helps people identify argumentative cues and logical fallacies; prebunking reliably improves short-term recognition of manipulative techniques; and fact-checking can correct targeted beliefs. Yet these gains rarely translate into durable resistance.

These interventions often assume that improving individuals' ability to detect errors will lead them to revise their judgments and ultimately form more accurate beliefs. This frames misinformation as a problem of truth detection. While normatively appealing, this is difficult to reconcile with how people actually process information—under uncertainty, limited resources, and with multiple goals. Thus, their ability to identify true from false claims depends less on the quality of their logical reasoning than on how well their confidence tracks a claim's plausibility given limited time, knowledge, and attention.

The theoretical and practical limitations of existing interventions are threefold. First, their benefits are tied to specific claims or tactics and decay as those evolve, making sustained impact at scale difficult. Second, because they build on a model of reasoning that emphasizes accuracy over the limits and practical goals that guide judgment, they leave blind spots. Third, regardless of individual capacities, information environments shape both what people attend to and how they assess plausibility. As platform incentives and group behavior distort the evidential landscape, even well-designed interventions may miss their mark when people's assessments rely on cues that no longer track evidential strength.

Because contemporary information environments distort the cues people rely on when assessing plausibility, interventions also need to support this assessment process rather than rely solely on binary truth detection. Strengthening metacognitive skills can help individuals monitor uncertainty and weigh competing cues. As these skills reinforce an internal process rather than relying on external arbiters of truth, they apply across domains and should be less exposed to partisan contestation. Metacognitive training still faces challenges such as scalability and interindividual variability, and evidence from complex, real-world settings remains limited. Its

implementation therefore requires careful empirical evaluation, yet it targets a dimension of judgment that current tools overlook, making it a valuable and complementary approach.

## The challenges of content-specific interventions

Interventions in contemporary information environments typically follow two routes: so-called prebunking aims to build resistance before exposure, and debunking seeks to correct beliefs after exposure. Prebunking is often implemented through psychological inoculation, which exposes users to weakened examples of manipulative techniques along with preemptive refutations. Its short-term effects are reliable in laboratory settings, but mid-to long-term benefits are highly contingent on reinforcements[1]. Debunking, by contrast, is retrospective and operates through follow-up fact-checking. When timely, accompanied by plausible alternatives, and issued by trusted sources, it can correct specific beliefs. Yet, corrections struggle with entrenched narratives, and can unintentionally broaden skepticism beyond the targeted claim[2].

A shared limitation is that both prebunking and debunking are content-specific: they target particular tactics or claims and must be continually updated as these evolve. Their effectiveness is therefore tied to the shifting information environment rather than to mechanisms that generalize across content, leaving the underlying process of judgment formation unchanged. As a result, sustained impact at scale is difficult. Lasting resistance is instead likely to come from pairing content-specific interventions with efforts to strengthen the cognitive processes through which judgments form under uncertainty.

## Reasoning is bounded and goal-directed

Prevailing approaches implicitly assume that, when encountering new information, individuals can attend to its relevant features, extract evidence in accordance with externally validated criteria of accuracy, and update their beliefs in proportion to the evidence. This treats reasoning as if it were a single-goal process aimed at maximizing truth, interpreting any departures from that aim as occasional errors attributed to bias or emotional interference. However, this view misrepresents both the *problem* people face and the *cognitive resources* they can realistically mobilize to solve it.

People rarely engage with information solely to assess its truth value. Rather than confronting a single, well-defined inference, individuals navigate overlapping problems under competing motivational and cognitive constraints. Accepting, rejecting, or further investigating a claim depends less on its objective accuracy[3] than on its anticipated usefulness for ongoing goals. This usefulness reflects a shifting balance between the need to refine one's understanding of the world and the expected practical, social, or emotional consequences of accepting or rejecting a claim. Reasoning thus involves selecting and attending to information that supports multiple priorities, not only evaluating its accuracy. Accordingly, people may

intentionally avoid or defer information when the psychological costs of knowing exceed its expected benefits[4].

Because cognition operates under limited resources, information is processed selectively. Attention and memory restrict what can be considered; people often ignore part of the available information to limit time and effort; and the way they take in and weigh available evidence is noisy, meaning that many evaluations are approximate rather than exhaustive. Whether limited by time or by computation, people must still decide when information appears plausible enough to act upon, and how to act.

Plausibility estimation is a continuous process through which people generate a graded sense of how likely a claim is to be right. It draws on whether the claim fits with existing knowledge, whether the source appears credible, and whether it aligns or conflicts with readily available facts. Because these assessments are triggered automatically when new information is encountered, plausibility estimation often functions as the earliest constraint on belief formation. But it also remains active whenever people engage in more deliberate reasoning about a claim. The binary choices that follow depend on this underlying estimate.

## Plausibility under distorted information environments

While plausibility estimation is a promising cognitive target, its reliability depends on the cues available in the environment, which are increasingly shaped by digital infrastructures. In digital environments, information spreads through systems optimized for engagement rather than the quality of evidence. Platform algorithms amplify material that is emotionally charged, aligned with group identity, and polarizing, making some claims appear more relevant or credible than warranted by evidence. Because plausibility estimation relies on assessments of coherence, credibility, and cue strength, algorithmically amplified cues are liable to distort the plausibility estimate itself. Attention is drawn toward visible indicators of engagement or social endorsement at the expense of cues that better track evidential strength. As a result, beliefs risk calibrating to these amplified cues rather than to the underlying evidence, stabilizing against later correction, and facilitating further spread.

Greater exposure to diverse or higher-quality information does not necessarily counteract these distortions, because the cues people rely on at the moment of judgment remain shaped by platform incentives. Across online and offline interactions, most individuals experience relatively balanced information regimes[5]. Yet, each platform's combination of incentives and social clustering still creates distinct information niches[6], making some claims disproportionately shaped by algorithmic pressures when they are encountered. In such contexts, the cues guiding plausibility become unreliable. Critical-thinking education can improve recognition of relevant cues over time, but plausibility estimation must also be calibrated for interpreting those cues and properly weighing available evidence.

## Strengthening plausibility estimation through metacognition

If misinformation persists not because people fail to reason, but because they reason adaptively under constraints, interventions must target the mechanisms that govern how judgments are formed. We propose that plausibility estimation is a promising but as yet insufficiently tested target for intervention.

When information is incomplete, ambiguous, or socially contested, people cannot verify truth directly. They instead rely on plausibility estimation: an assessment of how likely a claim is to be right given its fit with one's current understanding, available contextual cues, and the perceived credibility of its sources[7]. These checks provide an early filter through which new information passes, and continue to guide judgment as evidence accumulates. Resilience against misinformation, therefore, depends not only on these plausibility checks, but also on people's ability to monitor what they know, recognize when a claim is uncertain, notice when their confidence exceeds the strength of the evidence, and adjust it accordingly[8]. These metacognitive skills shape how beliefs relate to accumulated evidence and how different sources of information are weighed.

Near-perfect accuracy is unattainable under time and attention constraints. A more realistic goal is to attain proportionality between confidence in the evaluation of a claim and evidence. Reaching accuracy, therefore, requires good calibration, understood as how well confidence reflects plausibility in light of available evidence. Real-world decisions are typically binary: whether to vaccinate, share information, or enroll in a program. Error is minimized when such choices align with well-calibrated probabilities. Acting when plausibility is 55% rather than 45% may appear trivial, yet across many binary decisions, this small margin substantially reduces cumulative error. But, in complex information environments, confidence in judgments of truthfulness often becomes decorrelated from actual performance[9]. Part of this decoupling reflects that people often misestimate how much they understand: they confuse superficial knowledge or access to information with genuine understanding[10].

Metacognitive training typically focuses on improving probabilistic reasoning and calibration. It teaches people to compare a claim with relevant base rates, to revise their judgments as new evidence appears, to aggregate information from several independent cues, and to monitor signs of overconfidence. These components help individuals proportion their beliefs to the strength of the supporting evidence, thereby improving the reliability of plausibility estimation itself. Training programs developed in geopolitical forecasting and judgment literature[8,11,12] have produced measurable gains in estimation accuracy and reductions in systematic error.

Metacognitive approaches may nonetheless face familiar challenges in scalability, engagement, and long-term maintenance. While forecasting tournaments show that calibration skills can be improved through structured practice, whether comparable protocols can be adapted to misinformation contexts, generalize beyond training domains, and persist over time remain open empirical questions.

Aligning confidence with plausibility under uncertainty guards against premature certainty and indiscriminate doubt. Conversely, without this calibration, individuals may misallocate confidence, struggle with evaluating the strength of new claims, exhibit indiscriminate skepticism, or experience decision paralysis. Strengthening metacognitive calibration thus provides a psychologically grounded route to improving informational discernment and collective resilience.

## Rethinking the fight

Reframing reasoning as bounded and adaptive has significant implications for intervention design. By focusing on plausibility estimation and confidence regulation, metacognitive approaches target how judgments are formed under uncertainty. As advances in generative AI increasingly blur surface cues of authenticity, direct verification often becomes infeasible at the moment of judgment, increasing the importance of regulating confidence under uncertainty. Metacognitive approaches therefore complement content-specific approaches: the latter target the informational environment, whereas metacognition training targets the internal evaluation system through which all new information is filtered.

Even the most well-calibrated users must operate within informational environments where attention, not accuracy, determines reach, as digital architectures prioritize engagement over evidential quality. Under such conditions, the assumption that truth will prevail through open competition among ideas — the "marketplace of ideas" metaphor — breaks down. It

assumes that individuals seek truth and that greater access to information improves judgment, yet neither premise is supported empirically. On the other hand, it would be misguided to substitute the figure of the *perfectly rational agent* with its opposite caricature: the *perfectly biased agent* who must be corrected by controlled information systems. Ultimately, efforts should aim to design informational environments that are compatible with bounded and goal-directed reasoning.

Addressing misinformation at scale will require combining cognitive and structural approaches. Future research should test how different metacognitive strategies perform across populations and platforms, identifying moderators such as identity threat, trust calibration, and digital fatigue, and investigating how platform design can support, rather than constrain, discernment amid incentives for engagement. Critically, such work must assess whether gains in calibration are robust to misleading or sparse feedback, whether they generalize beyond the training domain, and whether they persist when confidence judgments are socially or emotionally incentivized. Efficacy should be assessed at two levels: individual calibration and accuracy improvements, and ecosystem diffusion quality. Such efforts demand collaboration across cognitive science, education, policy, and technology design. Future work should also assess the relative impact of content-based and metacognitive interventions, and determine how these approaches can be combined so that they align with the cognitive demands of the contexts in which they are deployed. Modeling work[13] shows that even if every misleading claim were identified and removed, aggregated belief accuracy would improve by only about 1%. This reflects the low prevalence of misinformation relative to the pervasiveness of uncertainty, which suggests that strengthening plausibility estimation may offer leverage in areas that content-based approaches cannot reach.

Metacognitive calibration is a critical and understudied lever for strengthening resilience in complex information ecosystems. The central challenge is not only that misinformation spreads, but that human judgments are formed under bounded resources and multiple goals, making the estimation of a claim's plausibility a primary cognitive bottleneck. Addressing misinformation, therefore, requires interventions that act on this evaluative process, rather than relying solely on binary truth detection. Metacognitive training targets this neglected dimension and thus represents one promising route for complementing content-level approaches in the fight against misinformation.

**Valentin Guigon** [iD][1,2] ✉**, Lucille Geay** [iD][3] **& Caroline J. Charpentier** [iD][1,2,4]

[1]Department of Psychology, University of Maryland, College Park, MD, USA. [2]Program in Neuroscience and Cognitive Science, University of Maryland, College Park, MD, USA. [3]Laboratoire CLLE (UMR 5263), Université Toulouse Jean Jaurès, Toulouse, France. [4]Brain and Behavior Institute, University of Maryland, College Park, MD, USA. ✉e-mail: vguigon@umd.edu

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

### Acknowledgements
The authors are responsible for the views expressed in this article and do not necessarily represent the views, decisions, or policies of the institutions with which they are affiliated.

### Author contributions
Valentin Guigon: Conceptualization, Writing - Original Draft, Writing - Review & Editing. Lucille Geay: Conceptualization, Writing - Original Draft, Writing - Review & Editing. Caroline J. Charpentier: Writing - Review & Editing.

### Competing interests
Caroline J. Charpentier is an Editorial Board Member of Communications Psychology, but was not involved in the editorial review of, or the decision to publish, this article. The other authors declare no competing interests.

### Additional information

**Peer review information** The manuscript was considered suitable for publication without further review at Communications Psychology. Primary Handling Editor: Marike Schiffer. A peer review file is available.

