## [Peer Review file · Communications Psychology]

Rethinking misinformation through plausibility estimation and confidence calibration

Corresponding Author: Dr Valentin Guigon

Version 0:

Decision Letter:

Dear Valentin,

Thank you for submitting your manuscript titled "Rethinking the Fight Against Disinformation" to Communications Psychology. I have given your manuscript careful consideration and the topic is of potential interest to the journal.

However, as I previously communicated via email, the piece is not one we can support as a Perspective or another long-form format. We would need it to be rewritten as a Comment. Comments have the following features:

- They're short and poignant, generally not longer than 1600-1800 words.
- They must make a compelling, novel argument.
- In style, they are opinion pieces, not review-type articles.
- The reference limit is 10.

A substantive part of your piece reviews well-known work, rather than focusing on your own proposal. The piece you have submitted is much closer to a mini-review -- it cites more than 25 references, providing an overview of the literature, including existing review content, rather than putting forward an original opinion. For example, it is not necessary to provide a lengthy introduction to bounded rationality -- bounded rationality is a well-established concept that has been reviewed elsewhere, including with reference to misinformation.

To be suitable for the format, the piece would need to undergo substantive edits. The review of the literature would need to be condensed into a few brief sentences that to provide the lay of the land. To become a forward-looking opinion piece, the ideas that are currently conveyed in the final paragraphs, especially those not fully fleshed out in the Conclusions, would need to take centre stage. Ideally, a Comment includes practical advice.

We would therefore like to invite you to revise your manuscript to address these concerns before we make a final determination on whether we can further pursue the piece.

I appreciate that the list of required changes is significant. If you instead choose to submit the present paper elsewhere, that is understandable - but please let me know so we can close your file.

Otherwise, we shall hope to receive your revised version as soon as you are able to complete the suggested revisions. If you anticipate a delay of more than three weeks, please let us know.

Thank you for your interest in Communications Psychology.

Best wishes,
Marika

Marika Schiffer, PhD
Chief Editor

Communications Psychology

Version 1:

Decision Letter:

** Please ensure you delete the link to your author homepage in this e-mail if you wish to forward it to your co-authors **

Dear Valentin,

Thank you for your patience during the editorial evaluation of your Comment titled "Rethinking the Fight Against Misinformation".

I much enjoyed reading the considerably improved manuscript, which is far better suited for the Comment format than the original submission. However, before we can make a final decision on its suitability for publication, we require some further revisions so that the piece may reach its full potential.

I have commented in detail on the manuscript (on the attached copy). As you will see, there are three areas of improvement in particular.

- 1) Conceptually, it is important that you clarify that you see the proposed intervention as a complementary strategy, not a replacement for existing interventions. This becomes clear in the outlook, but should narratively be introduced earlier on; in particular, it's important that the piece is not set up in such a way that the reader expects that your framework has demonstrably overcome all of the listed shortcomings of other interventions.
- 2) The piece is currently highly readable in the eyes of decision scientists, but too replete with framework-specific jargon to engage a very broad audience of psychology researchers. I appreciate the precision that comes with the use of formal terms. Nonetheless, given that many terms are mentioned only once in the text, and do not translate immediately into specific aspects of the proposed intervention, jargon needs to be considerably reduced.
- 3) As a final stylistic point, we recommend headings and subheadings that signal a position, not describe a process.

If the revised paper addresses all of these issues as well as the other requests listed in the document, we hope to be able to issue a final decision without further peer review.

Therefore, please review the changes in the attached copy of your manuscript, which has been edited for style, and address the comments and queries I have added. Please ensure you submit a clean and readable copy in revision.

Please use the following link to submit the above items:
Link Redacted

We hope to hear from you within four weeks; please let us know if the process may take longer.

Best regards,

Marika

Marika Schiffer, PhD
Chief Editor
Communications Psychology

Version 2:

Decision Letter:

** Please ensure you delete the link to your author homepage in this e-mail if you wish to forward it to your co-authors **

Dear Valentin,

Your revised manuscript titled "Rethinking the Fight Against Misinformation: from truth detection to plausibility evaluation" has undergone editorial evaluation. I am delighted to say that we are happy, in principle, to publish a suitably revised version in *Communications Psychology*.

If the revised paper is in *Communications Psychology* format, in an accessible style, and of appropriate length, we shall accept it for publication immediately.

EDITORIAL REQUESTS:

You will find a complete list of formatting requirements following this link: <https://www.nature.com/documents/commsj-style-formatting-checklist-comment.pdf>

Please use the checklist to prepare your manuscript for final submission. In the following, I also highlight some issues of particular importance.

** Title

Titles should be descriptive of the main message your manuscript conveys and should not exceed 90 characters (including spaces). Please note that punctuation is not allowed, nor are titles of the following format: "title: subtitle". Although the choice of title is largely yours, may I suggest the following:

Judging plausibility and gauging confidence to detect misinformation

** Preface

The paper's preface (up to 40 words; without references) should serve both as a general introduction to the topic, and highlight your position or proposal. Because we hope that researchers across all fields of psychology will be interested in your work, the preface should be as accessible as possible.

Democracies are vulnerable to misinformation. Prevailing interventions emphasize truth detection, but offer no panacea. Strengthening people's ability to evaluate the plausibility of information and calibrate their confidence under uncertainty offers a potential complementary route to misinformation detection.

SUBMISSION INFORMATION:

* *Communications Psychology* uses a transparent peer review system. On author request, confidential information and data can be removed from the published reviewer reports and rebuttal letters prior to publication. If you are concerned about the release of confidential data, please let us know specifically what information you would like to have removed. Please note that we cannot incorporate redactions for any other reasons.

*If you have not done so already, please alert me to any related manuscripts from your group that are under consideration or in press at other journals, or are being written up for submission to other journals (see www.nature.com/authors/editorial_policies/duplicate.html for details).

In order to accept your paper, we require the following:

* The final version of your text as a Word or TeX/LaTeX file, with any tables prepared using the Table menu in Word or the table environment in TeX/LaTeX and using the 'track changes' feature in Word.

At acceptance, you will be provided with instructions for completing the open access licence agreement on behalf of all authors. This grants us the necessary permissions to publish your paper. Additionally, you will be asked to declare that all required third party permissions have been obtained.

Please note that your paper cannot be sent for typesetting to our production team until we have received this information; **therefore, please ensure that you have this ready when submitting the final version of your manuscript.**

ORCID

Communications Psychology is committed to improving transparency in authorship. As part of our efforts in this direction, we are now requesting that all authors identified as 'corresponding author' create and link their Open Researcher and

Contributor Identifier (ORCID) with their account on the Manuscript Tracking System (MTS) prior to acceptance. ORCID helps the scientific community achieve unambiguous attribution of all scholarly contributions. For more information please visit <http://www.springernature.com/orcid>

For all corresponding authors listed on the manuscript, please follow the instructions in the link below to link your ORCID to your account on our MTS before submitting the final version of the manuscript. If you do not yet have an ORCID you will be able to create one in minutes.

IMPORTANT: All authors identified as 'corresponding author' on the manuscript must follow these instructions. Non-corresponding authors do not have to link their ORCIDs but are encouraged to do so. Please note that it will not be possible to add/modify ORCIDs at proof. Thus, if they wish to have their ORCID added to the paper they must also follow the above procedure prior to acceptance.

To support ORCID's aims, we only allow a single ORCID identifier to be attached to one account. If you have any issues attaching an ORCID identifier to your MTS account, please contact the [Platform Support Helpdesk](http://platformsupport.nature.com/).

Link Redacted

We hope to hear from you within two weeks; please let us know if the process may take longer.

Best regards,

Marika

Marika Schiffer, PhD
Chief Editor
Communications Psychology

**** Visit Nature Research's author and referees' website at www.nature.com/authors for information about policies, services and author benefits****
